# Quantitative assessment of protein activity in orphan tissues and single cells using the metaVIPER algorithm

Hongxu Ding[1,2], Eugene F. DouglassJr.[1], Adam M. Sonabend[3], Angeliki Mela[3], Sayantan Bose[1,9], Christian Gonzalez[1,10], Peter D. Canoll[3], Peter A. Sims[1], Mariano J. Alvarez[1,4] & Andrea Califano[1,4,5,6,7,8]

We and others have shown that transition and maintenance of biological states is controlled by master regulator proteins, which can be inferred by interrogating tissue-specific regulatory models (interactomes) with transcriptional signatures, using the VIPER algorithm. Yet, some tissues may lack molecular profiles necessary for interactome inference (orphan tissues), or, as for single cells isolated from heterogeneous samples, their tissue context may be undetermined. To address this problem, we introduce metaVIPER, an algorithm designed to assess protein activity in tissue-independent fashion by integrative analysis of multiple, non-tissue-matched interactomes. This assumes that transcriptional targets of each protein will be recapitulated by one or more available interactomes. We confirm the algorithm's value in assessing protein dysregulation induced by somatic mutations, as well as in assessing protein activity in orphan tissues and, most critically, in single cells, thus allowing transformation of noisy and potentially biased RNA-Seq signatures into reproducible protein-activity signatures.

[1] Department of Systems Biology, Columbia University, New York, NY 10032, USA. [2] Department of Biological Sciences, Columbia University, New York, NY 10027, USA. [3] Department of Pathology and Cell Biology, Columbia University, New York, NY 10032, USA. [4] DarwinHealth Inc, New York, NY 10032, USA. [5] Herbert Irving Comprehensive Cancer Center, Columbia University, New York, NY 10032, USA. [6] J.P. Sulzberger Columbia Genome Center, Columbia University, New York, NY 10032, USA. [7] Department of Biomedical Informatics, Columbia University, New York, NY 10032, USA. [8] Department of Biochemistry and Molecular Biophysics, Columbia University, New York, NY 10032, USA. [9] Present address: GlaxoSmithKline, King of Prussia, PA 19406, USA. [10] Present address: Amsterdam Neuroscience, Amsterdam 1081, The Netherlands. Correspondence and requests for materials should be addressed to M.J.A. (email: malvarez@darwinhealth.com) or to A.C. (email: andrea.califano@columbia.edu)

Most biological events are characterized by the transition between two cellular states representing either two stable physiologic conditions, such as during lineage specification[1,2] or a physiological and a pathological one, such as during tumorigenesis[3,4]. In either case, cell state transitions are initiated by a coordinated change in the activity of key regulatory proteins, typically organized into highly interconnected and auto-regulated modules, which are ultimately responsible for the maintenance of a stable endpoint state. We have used the term "master regulator" (MR) to refer to the specific proteins, whose concerted activity is necessary and sufficient to implement a given cell state transition[5]. Critically, individual MR proteins can be systematically elucidated by computational analysis of regulatory models (interactomes) using MARINa (Master Regulator Inference algorithm)[6] and its most recent implementation supporting individual sample analysis, VIPER (Virtual Inference of Protein activity by Enriched Regulon)[7]. These algorithms prioritize the proteins representing the most direct mechanistic regulators of a cell state transition, by assessing the enrichment of their transcriptional targets in genes that are differentially expressed. For instance, a protein would be considered significantly activated in a cell-state transition if its positively regulated and repressed targets were significantly enriched in overexpressed and under-expressed genes, respectively. The opposite would, of course, be the case for an inactivated protein. As proposed in[7], this enrichment can be effectively quantitated as Normalized Enrichment Score (NES) using the Kolmogorov–Smirnov statistics[8]. We have shown that the NES can then be effectively used as a proxy for the differential activity of a specific protein[7]. Critically, such an approach requires accurate and comprehensive assessment of protein transcriptional targets. This can be accomplished using reverse-engineering algorithms, such as ARACNe[9] (Accurate Reverse Engineering of Cellular Networks) and others (reviewed in ref. [10]), as also discussed in ref. [7].

MARINa and VIPER have helped elucidate MR proteins for a variety of tumor related[11–17], neurodegenerative[18–20], stem cell[21,22], developmental[6], and neurobehavioral[23] phenotypes that have been experimentally validated. The dependency of this algorithm on availability of tissue-specific models, however, constitutes a significant limitation because use of non-tissue-matched interactomes severely compromises algorithm performance[11]. Since ARACNe requires $N \geq 100$ tissue-specific gene expression profiles, representing statistically independent samples, some tissue contexts may lack adequate data for accurate interactome inference. These "orphan tissues" include, for instance, rare or poorly characterized cancers, as well as progenitor states during lineage differentiation. In addition, the specific tissue lineage of a sample may be poorly defined, thus preventing selection of appropriate interactome models. Consider, for instance, a single cell isolated from a heterogeneous sample, such as whole brain or stroma-infiltrated tumor, where many highly distinct and often uncharacterized cell lineages are inextricably commingled.

To address this challenge, we reasoned that while regulatory models are clearly lineage specific, due to the distinct epigenetic state of the cells, the transcriptional targets of a specific protein (i.e., its regulon) may be at least partially conserved across a small subset of distinct lineages. Thus, once a sufficient number of tissue-specific interactomes is available, the likelihood that one or more of them may represent a good model for the regulon of a specific protein increases, even though one may not know a priori which model may represent the best match for each protein. Indeed, the regulons of different proteins may be optimally represented within different interactomes. This is further helped by the fact that VIPER analysis is robust if at least 40% of a protein's regulon is accurately inferred[7]. Conversely, as shown in Supplementary Fig. 1 when a protein regulon is incorrectly assessed for a specific tissue, it is not consistent with the tissue-specific gene expression signature, thus producing no significant enrichment. Taken together, these observations constitute the basis for the implementation of a context-independent algorithm for protein activity assessment (metaVIPER).

MetaVIPER implements a statistical framework for evidence integration across a large repertoire of context-specific interactomes, see Methods for details. The algorithm is based on the assumption that only regulons that accurately represent the transcriptional targets of specific proteins in the tissue of interest will produce statistically significant enrichment in genes that are differentially expressed in that tissue (Fig. 1a).

To assess whether metaVIPER can effectively assess protein activity in context-independent fashion we perform a number of distinct benchmarks. First, we assessed whether results produced by analysis of context-specific interactomes (e.g., inferred from breast cancer samples) could be effectively reproduced when only interactomes from other tissues are used in integrative fashion. We also test whether the ability to assess dysregulation of proteins whose encoding gene harbored a recurrent somatic alteration was improved by metaVIPER. Finally, we assess the algorithm's ability to transform low-depth single cell RNA-Seq (scRNA-Seq) profiles into highly reproducible protein activity profiles that accurately reflect cell state, while removing technical artifacts and batch effects, compared to state of the art gene expression based methods. These improvements significantly increase the ability to analyze the biological function and relevance of gene products whose mRNAs are undetectable in low-depth, scRNA-Seq data (dropout effect), without any a priori knowledge of the single cell's lineage. In particular, it allows more stringent analysis of critical lineage markers, for which no mRNA reads may be detectable in individual cells, either individually or as a set, supporting a "virtual FACS" analysis.

## Results

**Overview of metaVIPER.** Let us assume a tissue context T for which a matched tissue-specific interactome was not available. Furthermore, without loss of generality, let us focus on a specific protein of interest P and on its T-specific regulon $R_T$. Given a sufficient number of additional tissues $T_1 \ldots T_N$ for which accurate, context-specific interactomes are available, we hypothesize that $R_T$ will be at least partially recapitulated in one or more of them. Based on previous results[7], VIPER can accurately infer differential protein activity, as long as 40% or more of its transcriptional targets are correctly identified. As a result, even partial regulon overlap may suffice. Indeed, paradoxically, there are cases where a protein's regulon may be more accurately represented in a non-tissue matched interactome than in the tissue-specific one. This may occur, for instance, when expression of the gene encoding for the protein of interest has little variability in the tissue of interest and greater variability in a distinct tissue context where the targets are relatively well conserved. A key challenge, however, is that one does not know a priori which of the tissue-specific interactomes may provide reasonable vs. poor models for $R_T$.

To address this challenge, we leverage previous studies showing that if an interactome-specific regulon provides poor $R_T$ representation, approaching random selection in the limit, then it will also not be statistically significantly enriched in genes that are differentially expressed in a tissue-specific signature $S_T$. Thus, if one were to compute the enrichment of all available regulons for the protein P in the signature $S_T$, only those providing a good representation will produce statistically significant enrichment, if P is differentially active in the tissue of interest. Conversely, if the

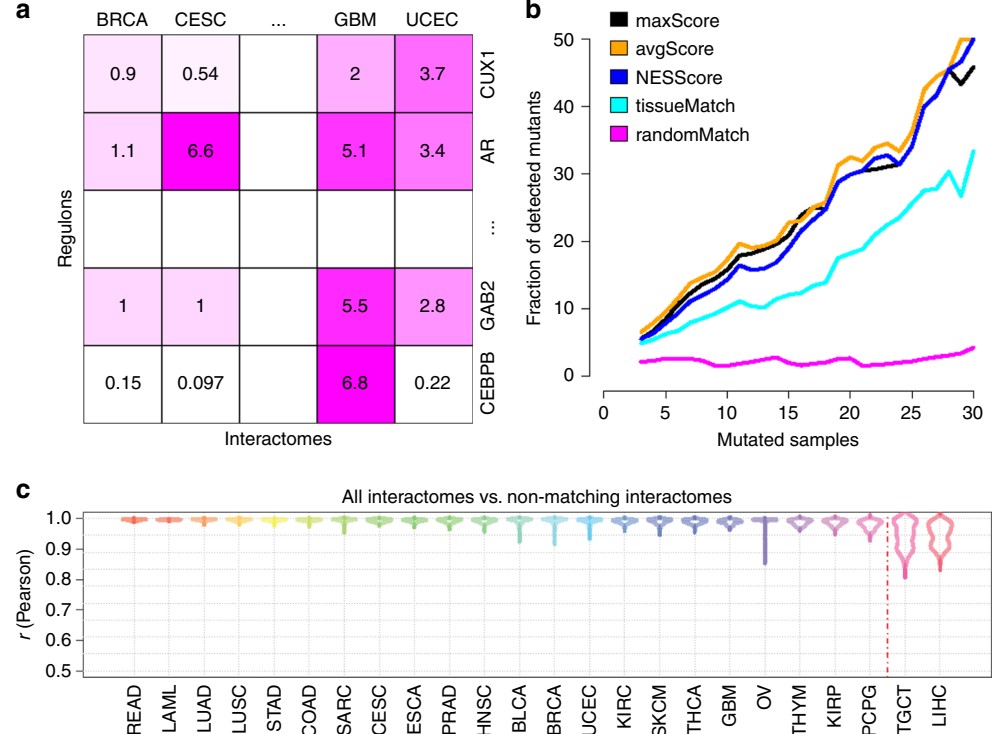

**Fig. 1** Inferring protein activity with metaVIPER. **a** Overview of metaVIPER. The set of transcriptional targets for each regulatory protein (its regulon) constitutes the fundamental building blocks of an interactome, which reflect its overall, context-specific regulatory control structure. MetaVIPER identifies the regulon that best recapitulates the regulatory targets of a protein by assessing its enrichment in the tissue-specific differential expression signature. In the example shown here, for instance, the regulon for protein CUX1 in an unknown or orphan tissue is better recapitulated by the uterine corpus endometrial carcinoma (UCEC)-based regulon, while the transcriptional program for the androgen receptor protein (AR) is better recapitulated by the cervical squamous cell carcinoma and endocervical adenocarcinoma (CESC) and glioblastoma (GBM)-based regulons. The numbers indicate −log10($p$-value) for enrichment of the regulons on the gene expression signature, as computed by VIPER. **b** Impact of recurrent coding somatic mutations on metaVIPER-inferred protein activity. Fraction of proteins showing significant association between metaVIPER-inferred protein activity and somatic mutations ($p < 0.01$) is presented. VIPER analysis was performed using the tissue-matched network (tissueMatch), metaVIPER was performed by integrating the results from individual interactomes using maxScore, avgScore, and NESScore methods; the baseline control was computed by using intercatomes selected at random (randomMatch). The $X$-axis represents the minimum number of TCGA samples presenting the specific gene mutation required for inclusion of the encoded protein in the analysis. **c** Inference of protein activity for orphan tissues. MetaVIPER can effectively reproduce differential protein activity in TCGA tissues, even when the corresponding matched interactome is removed from the analysis. The only partial exception is represented by two tissue lineages—liver hepatocellular carcinoma (LIHC) and testicular germ cell tumors (TGCT)—which are defined by highly specific regulatory programs. The probability density distribution for the correlation between protein activities (NES) inferred by metaVIPER using all available interactomes vs. metaVIPER using all, but the tissue-matched interactome (Pearson's correlation) across all samples is shown by the violin plots

protein is not differentially active in T, then no regulon $R_{T1}$ … $R_{TN}$ should produce statistically significant enrichment. If these assumptions were correct, given a sufficient number of tissue-specific interactomes, this would provide an efficient way to integrate across them to compute the differential activity of arbitrary proteins in tissue contexts for which a suitable interactome model may be missing.

To determine the best strategy for integrating the statistics of the enrichment across multiple interactomes, we compared several approaches. Specifically, for each protein, we first computed enrichment using a tissue-matched interactome (tissueMatch). This corresponds to the original implementation of the VIPER algorithm. We then compared these results to those obtained using different metrics to integrate across the regulons of all non-tissue-matched interactomes, including (a) the NES with the most statistically significant absolute value (maxScore), (b) the average of all NES scores (avgScore), and (c) the weighted-average of all NES scores, weighed by the NES absolute value (NESScore). For these tests, we used a total of 24 interactomes generated from TCGA cohorts, see Supplementary Table[24].

To objectively evaluate the performance of these alternative integrative methods, we considered a comprehensive set of proteins, whose genes harbor recurrent somatic mutations, as reported by both TCGA and COSMIC (see Methods). These mutations drive tumorigenesis by altering the activity of key oncogenes and tumor suppressors and have been used to identify proteins for targeted inhibitors, based on the oncogene additional paradigm[25]. We thus assessed method performance by assessing the statistical significance of the correlation between metaVIPER-inferred protein activity and the presence of a recurrent genetic alterations in the corresponding gene locus ($p < 0.01$), under the assumption that better methods would yield higher significance. To produce an optimal metric across all recurrent mutational events, we assessed correlation as a function of recurrence (Fig. 1b). Indeed, the more recurrent a mutation is, the more likely it is to be functionally relevant and thus affect the corresponding protein's activity. Recurrence is reported as the number of samples in TCGA and COSMIC where a specific gene locus was mutated, see Methods. As shown in Fig. 1b, there is a clear trend showing that the more recurrently mutated a gene

locus is, the larger the fraction of proteins showing statistically significant correlation between metaVIPER-inferred protein activity and mutational state. For instance, about 50% of the genes harboring locus-specific mutations in at least 30 TCGA samples could be detected as producing differentially active proteins by metaVIPER analysis ($p < 0.01$).

Surprisingly, based on this metric, all four strategies for cross-tissue integration (metaVIPER) significantly outperformed the use of tissue-specific interactomes, i.e., the original VIPER algorithm (tissueMatch). This suggests that integrating the structure of regulatory networks across a large number of representative tissue types provides a more informative regulon representation on an individual protein basis. The randomMatch method serves as a baseline negative control, in which for each sample, protein activity was computed using VIPER with an interactome selected at random. As discussed in the following sections, we performed several additional benchmarks to comprehensively and systematically assess the method's performance in orphan tissues, as well in single cells.

**MetaVIPER-based protein activity inference in orphan tissues.** Small sample size severely undermines the performance of ARACNe, which typically requires at least 100 independent samples, representative of the same tissue lineage[26] to perform accurate regulon inference for VIPER analysis. This significantly limits the ability to accurately measure protein activity in orphan tissues, defined as rare or poorly characterized tissue types, for which the number of available gene expression profiles is not sufficient to produce an accurate interactome model. For instance, considering tumor cohorts in the TCGA repository, we identified Cholangiocarcinoma ($N = 36$) and Uterine Carcinosarcoma ($N = 57$) could be considered orphan tissues for which an accurate ARACNe network could not be generated. Orphan tissues also include a variety of normal or non-cancer, disease-related cell states that lack appropriate gene expression profile characterization, including many of the intermediate states of differentiation representing multipotent or progenitor population.

Since metaVIPER is designed to infer protein activity without requiring a tissue-specific regulatory model, we designed an objective benchmark to assess metaVIPER's ability to accurately measure protein activity in orphan tissues. We first assembled a gold-standard set using metaVIPER to assess the activity of all proteins for which an ARACNe regulon was generated (see Methods), in each sample of each TCGA cohort, using all available TCGA interactomes including the tissue-matched one. This is preferred to using only the tissue-matched interactome because from the objective benchmark using mutational data this methodology has emerged as being more accurate than the original VIPER analysis. However, for completeness, we also report results of this analysis using the tissue-matched interactomes as gold-standard, see Supplementary Fig. 2. We then performed the same analysis using metaVIPER with all available TCGA interactomes, except for the tissue-matched one. For instance, consider Rectum Adenocarcinoma (READ) as a tumor for which an ARACNe interactome could not be accurately inferred. We would then compute the VIPER-inferred activity of all proteins in each TCGA READ sample using either all available TCGA interactomes (gold-standard reference) or all interactomes except for the READ interactome, exactly as if it were not available. We then measure overall protein activity correlation between the two analyses as a quality metric for metaVIPER ability to correctly infer protein activity in the absence of a tissue-matched interactome. This benchmark was performed for each of the all 24 tissue types in TCGA, see Supplementary Table[24].

Results show extremely strong average correlation ($\rho > 0.97$) between the two analyses for 22 out of 24 tissues (excluding liver hepatocellular carcinoma (LIHC) and testicular germ cell tumors (TGCT)). This suggests that, even in the absence of a tissue-matched model, most tissues may be studied virtually without loss of resolution using metaVIPER (Fig. 1c, Supplementary Fig. 2). Thus most orphan tissues can be studied using metaVIPER with virtually no notable result quality degradation. Not surprisingly, the two outlier tissues have a rather unique nature. Indeed, LIHC is originated from hepatocytes, which are unique endoderm derived secretory cells[27]. Similarly, TGCT originate from testicular germ cells, which are specialized pluripotent cells that give rise to gametes[28]. Hepatocytes and testicular germ cells are thus highly specialized tissues with no other related tissues among the 24 in TCGA. However, as the number of interactomes in our repertoire grows the probability of having true outlier tissues will decrease. Note, however that, despite their specialized nature even the two outlier tissues presented high average correlation with the results of the tissue-matched analysis ($\rho > 0.95$).

This raises the important issue of an objective metric to assess whether metaVIPER—when used with a specific repertoire of tissue-specific interactomes—is adequate for inferring protein activity in tissues lacking a matched interactome (i.e., orphan tissues). To achieve this goal, as proposed in ref.[9], we will use the Empirical Cumulative Distribution Function of the absolute value of the VIPER NES ($ECDF_{|NES|}$) of all proteins in an orphan tissue sample or samples[11]. In Supplementary Fig. 7, we show violin plots for the $ECDF_{|NES|}$ of each TCGA cohort, using the corresponding tissue-matched interactome. The rightmost plot (TCGA) shows the average of all cohort-specific probability densities. This provides a useful reference to assess whether a specific interactome repertoire is adequate for the metaVIPER-based analysis of an orphan tissue. For instance, we analyzed LAML samples using only a GBM interactome, which would be clearly inappropriate since LAML and GBM cells belong to epigenetically distinct lineages. The result is shown in the first-to-last violin plot (Neg.Ctrl.). As shown this ECDF is clearly an outlier with respect to All-TCGA. Thus, by comparing the ECDF for a tissue of interest against the All TCGA reference, one can effectively assess the quality of the analysis.

**Single cell analysis.** The last few years have seen tremendous development of single cell profiling methodologies and in particular of scRNA-Seq. The advent of these technologies provides new insight in understanding transition, maintenance, and cell–cell communication processes, across cell states and at an individual cell resolution[29]. However, a major challenge of these approaches is related to the very low depth of sequencing ranging between 10 and 200K reads per cell. While this is sufficient to perform coarse analyses, such as multi-dimensional clustering to identify molecularly distinct sub-populations, it is extremely ineffective in precisely quantitating the expression of individual genes. Indeed, the vast majority of genes lack even one mRNA read in individual cells (dropouts) and a large number have a single read. Due to these significant dropout effects, elucidating biological mechanisms at the single cell level remains challenging. In contrast, as shown in ref.[7], VIPER analysis is largely unaffected by sequencing depth because differential protein activity is assessed based on the differential expression of hundreds of transcriptional targets. Thus, measurement and biological noise sources are effectively averaged out, resulting in highly reproducible measurements. Indeed, we have shown that VIPER-inferred protein activity profiles from FFPE samples were extremely well correlated to those from fresh-frozen samples, despite

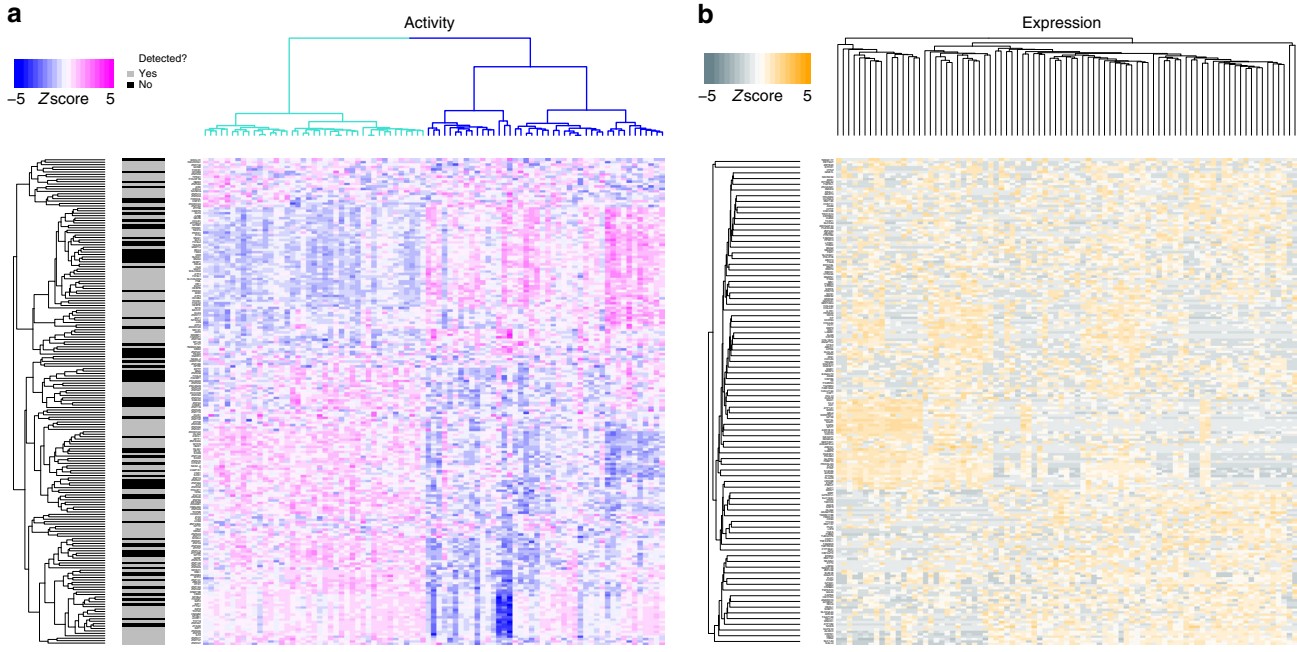

**Fig. 2** Inference of protein activity for single cells from GBM mouse model. **a** MetaVIPER-based protein activity analysis of single cells from a mouse GBM model[27,28] by unsupervised clustering using all annotated transcriptional factors, co-transcriptional factors, and signaling proteins. Two major clusters were identified, corresponding to established mesenchymal (MES, blue) and proneural (PN, turquoise) subtypes, with varying proliferative (Prolif) potential[11]. Indeed, among the top 200 transcriptional factors (i.e., with the highest inter-cluster activity variability), we found established master regulatory transcriptional factors of the MES (*FOSL1, FOSL2, RUNX1, CEBPB, CEBPD, MYCN, ELF4*), PN (*OLIG2, ZNF217*), and Prolif (*HMGB2, SMAD4, PTTG1, E2F1, E2F8, FOXM1*) subtypes[13]. **b** Subtype representation is lost when clustering is performed based on gene expression profiles

dramatic loss of correlation at the gene expression level[7], leading to NYS CLIA approval of two VIPER-based tests. As a result, one would expect VIPER to be well suited to performing analysis of single cell populations in a way that is amenable to quantitative protein activity assessment.

Unfortunately, however, when dealing with heterogeneous samples, the specific tissue context of each individual cell cannot be determined a priori. Even if this were possible, it is unlikely that context specific interactomes would be available for rare lineages and progenitor states that are captured by single cell profiling methodologies. MetaVIPER represents a useful alternative in these cases, because, while preserving the robustness of VIPER, it is agnostic to tissue type and should thus be well suited to analysis of single cell gene expression profiles from heterogeneous tissues.

To illustrate metaVIPER applicability to single cell expression profile data, we specifically profiled 85 single cells (see Methods) from a mouse glioblastoma (GBM) model[30,31]. Previous studies have demonstrated that GBM comprises two major subtypes, mesenchymal (MES) and Proneural (PN), which may present different proliferation capability (Prolif)[13,32–34]. We inferred protein activity at the single cell level by metaVIPER analysis across 5 brain tumor interactomes, and 24 TCGA human cancer tissue interactomes (see Methods and Supplementary Table). Contrary to gene expression profile analysis, the inferred protein activity signatures clearly captured single cells representing MES and PN subtypes. Indeed, unsupervised metaVIPER analysis recapitulated previously reported subtype-specific MR proteins[13], which were identified among the most dysregulated on a single cell basis (Fig. 2a). Such level of resolution could not be recapitulated by differential gene expression analysis, largely due to transcript-level noise in individual cells (Fig. 2b). Unsupervised clustering analysis of metaVIPER-inferred protein

activity efficiently separated single cells in two major groups, with ~40% of the cells recapitulating the activity pattern of previously described MR proteins of MES (*FOSL1, FOSL2, RUNX1, CEBPB, CEBPD, MYCN, ELF4*), and the remaining ~60% recapitulating those of the PN, such as *OLIG2* and *ZNF217*. In sharp contrast, unsupervised, gene expression based cluster analysis could not effectively separate individual cells in distinct clusters (Fig. 2a, b). Indeed, ~40% of the critical subtype-related proteins were undetectable at the gene expression level in any of the single cells (black horizontal bars in Fig. 2a). Expression profiles from single cells are very noisy, due to low sequencing depth, thus reducing the ability to study their biology. Indeed, low depth of sequencing represents a major confounding factor that can be effectively remedied by metaVIPER analysis.

Quality of single cell gene expression profiles is generally reflected by the number of detected genes[29]. Higher quality gene expression profiles, as identified by higher transcriptome complexity, tend to result in higher correlation between the profiles of single cells in the same sub-population clusters (Supplementary Fig. 3A, B). Once processed with metaVIPER, however, not only intra-population correlation between individual cells increases significantly but it also becomes virtually independent of transcriptome complexity (Supplementary Fig. 3C). This is because protein activity inference is based on the expression of many target genes and is thus much more robust than estimating gene expression from a single measurement, thus improving resilience to low-quality data.

We further tested our methodology on single cell data from tissue representing a complex mixture of melanoma cells and infiltrating B and T lymphocytes[35]. By integrating interactomes representative of skin cutaneous melanoma (SKCM, see Methods), B[9] and T[36] lymphocytes, as well as 24 TCGA human cancer tissue[24] (Supplementary Table), metaVIPER was able to infer

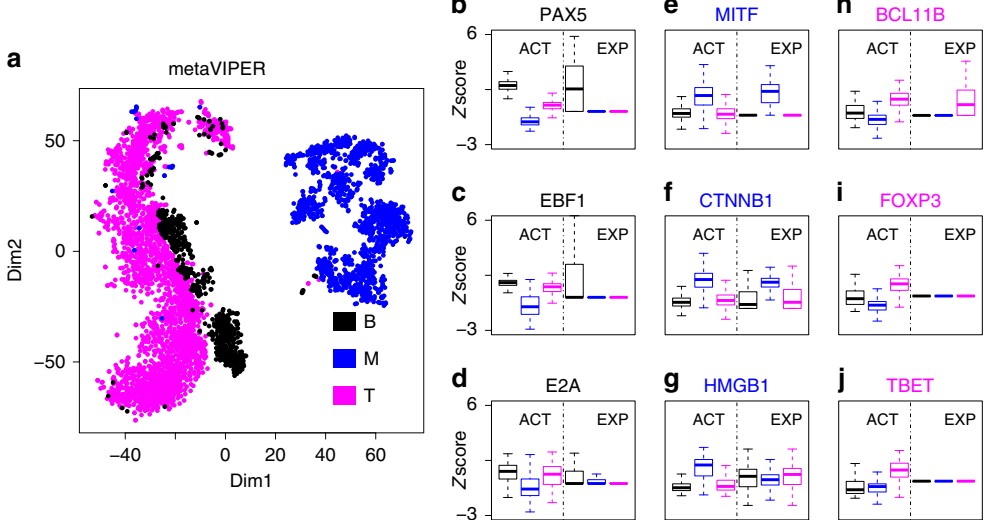

**Fig. 3** Inference of protein activity for single cells profiled by Tirosh et al.[35]. **a** Annotated cell types (B: B lymphocyte, T: T lymphocyte, M: melanoma cell) were separated by t-SNE analysis, using metaVIPER-inferred activity for all annotated transcriptional factors, co-transcriptional factors, and signaling proteins. Boxplots show metaVIPER-inferred activity, as well as gene expression for tissue-specific lineage markers, including *PAX5*[37], *EBF1*[38], and *E2A*[39] for B lymphocyte (**b**–**d**), *MITF*[40], *CTNNB1*[41], and *HMGB1*[42] for melanocyte (**e**–**g**), *BCL11B*[43], *FOXP3*[44], and *TBET*[45] for T lymphocyte (**h**–**j**). While these markers are significantly differentially active in these tissues, they could not be effectively assessed at the single cell level, either because no mRNA reads were detected or because markers were not statistically significant in terms of differential gene expression. Boxplots showed the median, lower/upper whiskers, and hinges of z-scores

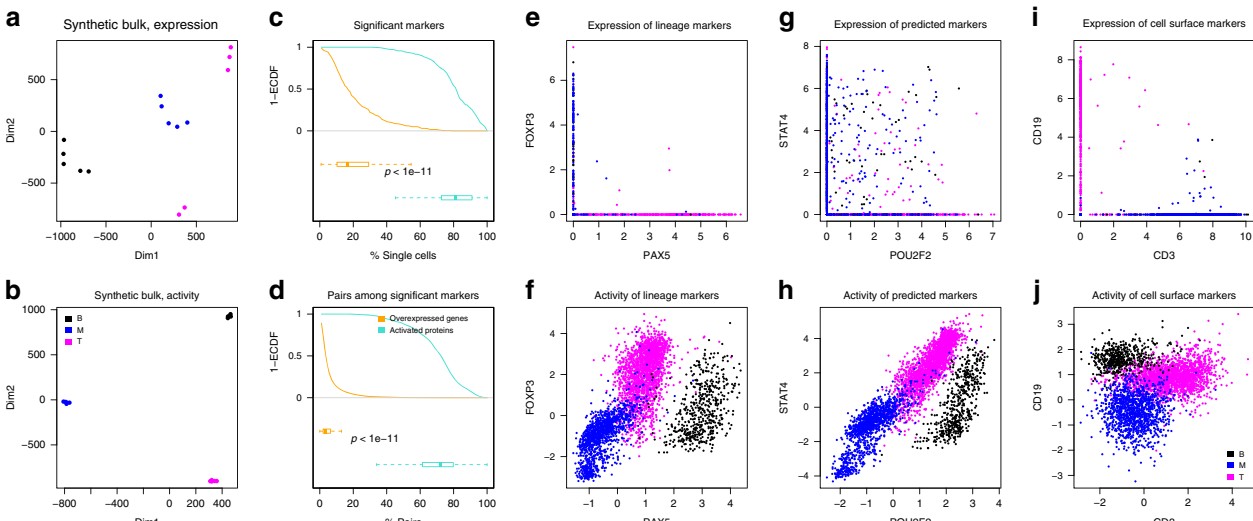

**Fig. 4** Comparative analysis of single cell metaVIPER performance compared to gene expression based methods. We identified the 100 most differentially expressed genes and differentially active proteins based on the analysis of five synthetic bulk samples created by averaging the expression of 100 randomly selected single cells from the melanoma, B cell, and T cell population clusters, respectively. **a**, **b** Based on t-SNE analysis, synthetic bulk samples clustered more tightly when analyzed based on VIPER-inferred protein activity than based on gene expression. **c** This panel shows the percent of the top 100 most differentially expressed genes/active proteins recapitulated as significantly differentially expressed/active in a given fraction of individual cells against the average expression/activity in a distinct cluster (e.g., a T cell vs. the average of all B cells). The yellow and turquoise curves (1-ECDF) and boxplots (median, lower/upper whiskers, and hinges) summarized the results of RSEM and metaVIPER-based analyses, respectively. **d** The same analyses were repeated to assess reproducible differential expression/activity of a gene/protein pair, as relevant for virtual FACS analyses. **e**, **f** Virtual FACS analyses using expression and activity of established lineage marker TFs by RSEM and metaVIPER-based analysis (see main text and Fig. 3 for details). **g**, **h** Virtual FACS analysis using expression and activity of *STAT4* and *POU2F*—both identified as differentially expressed and active candidate biomarkers from bulk sample analyses—using the same methods. **i**, **j** Virtual FACS analysis based on expression and activity of CD3 and CD19 cell surface markers, as used in standard FACS analyses, using the same methods

protein activity profiles that effectively discriminate between these different cell types. Furthermore, it revealed differential activity of established lineage markers that could not be detected at the gene expression level (Fig. 3b–j). This represents a critical value of this approach, as many important lineage markers and other transcriptional regulators may yield no scRNA-Seq reads, due to their relatively low transcript abundance combined with low sequencing depth. Based on a metric assessing the dynamic range of protein activity in different sub-clusters, metaVIPER significantly outperformed single-regulon-based VIPER analysis on this dataset (Supplementary Fig. 4). Most importantly, metaVIPER correctly inferred the differential, tissue-specific activity of established lineage determinants at the single cell level (Fig. 3b–j).

For instance, PAX5[37], EBF1[38], and E2A[39] showed significantly higher activity in B lymphocytes (one-tail, $p < 10^{-10}$); MITF[40], CTNNB1[41], and HMGB1[42] showed significantly higher activity in melanoma cells (one-tail, $p < 10^{-10}$); finally, BCL11B[43], FOXP3[44], and TBET[45] showed significantly higher activity in T lymphocytes (one-tail, $p < 10^{-10}$). Conversely, we could not detect significant gene expression differences for most of these genes (e.g., $p_{HMGB1} > 0.9$) in melanoma cells, or expression was barely detected at all (average transcripts per million < 1), see E2A in B lymphocytes or FOXP3 and TBET in T lymphocytes, for instance (Fig. 3b–j).

To provide a more systematic comparison of the improvements offered by metaVIPER analysis of single cells against approaches based on state-of-the-art gene expression analysis algorithms, using the same mixture of T, B, and melanoma cells described in the previous section. Most methods designed to address the gene dropout issue in scRNA-Seq profiles are not intended to perform differential expression analysis of two individual cells but rather only of single cell subsets representing molecularly distinct clusters/subtypes[46–48]. To perform this analysis, we thus quantified single cell gene expression using RSEM[49], which pre-assembles sequencing reads into transcripts, thus providing more accurate single cell gene expression quantification[50]. We then assessed the fraction of single cell pairs from two distinct clusters (e.g., B and T cell related) that could recapitulate differentially expressed genes and differentially active proteins, as originally detected from their corresponding bulk cell populations. For each cluster, we generated "synthetic bulk" expression profiles by averaging 100 randomly selected single cells, based on which we generated "synthetic bulk" protein activity profiles. As shown in the corresponding t-SNE plots, synthetic bulk profiles from metaVIPER-inferred protein activity analysis (Fig. 4b) were much tighter than those produced by gene expression analysis (Fig. 4a), suggesting that VIPER-inferred protein activity is more reproducible across samples than mRNA expression. Finally, we assessed the fraction of the 100 most differentially expressed genes and differentially active proteins (as assessed from bulk sample analysis) that could be recapitulated in a given fraction of single cells when compared to the bulk expression of a different cluster (e.g., a single T-cell vs. all cells in the melanoma cluster). As shown in Fig. 4c, differential activity (turquoise curve) significantly outperformed RSEM-based differential gene expression analysis (yellow curve). This becomes even more evident when considering pairs of differentially expressed genes or active proteins (e.g., gene X and Y being both differentially expressed in a single cell if they are both differentially expressed in the bulk) (Fig. 4d). The latter is important as it supports use of metaVIPER to generate analyses similar to what is normally accomplished by FACS, using two or more markers, using any of the ~6000 proteins assessed by the algorithm not limited by antibody availability. This is shown in Fig. 4e–j, where virtual FACS plots are shown for critical lineage markers of these populations using gene expression (top plots) or protein activity (bottom plots). As shown, it is virtually impossible to identify cell clusters based on

selected marker pairs at the gene expression level. Indeed, most of the cells are found either on the x-axis (no detectable expression of the Y-marker) or on the y-axis (no detectable expression of the X-marker) or at the intersection of the two axes (no detectable expression of either marker). In contrast, metaVIPER analysis generates virtual FACS plots that are consistent with what would be produced by an actual FACS assay. For instance, consider CD19 and CD3, which are classic B and T cells markers, respectively. From metaVIPER analysis (Fig. 4j), one can clearly identify a CD19+/CD3− cluster corresponding to B cells, a CD19−/CD3+ cluster corresponding to T cells, and a CD19−/CD3− cluster corresponding to melanoma cells. Yet, this is not possible when considering single cell gene expression (Fig. 4i).

Finally an additional value of the algorithm is that processes that are not consistent with the transcriptional regulatory architecture of the cells of interest are effectively filtered out by the interactome analysis. This is useful, for instance, in eliminating bias due to different chemistry of single cell profiling or batch effects due to use of different gene expression quantification methodologies (Supplementary Fig. 5 and 6). This is helpful as these biases and batch effects represent a major obstacle to the integrative analysis of gene expression data generated in different labs or using slightly different reagent batches.

Taken together, these data show that metaVIPER represents a useful methodology for the analysis of single cell data and, in particular, for the identification of lineage-specific regulatory programs and lineage markers in samples comprising a heterogeneous mixture of single cells.

## Discussion

We have shown that integration of multiple interactomes using an evidence integration platform (metaVIPER) can provide accurate assessment of protein activity independent of tissue lineage. By systematic, we mean that activity of 6000 proteins can be reproducibly assessed from any tissue, independent of their gene expression; this is especially valuable in single cell analyses. MetaVIPER can thus help infer activity of key regulators in tissues lacking a matched interactome—either due to low sample availability (orphan tissues) or to lack of tissue lineage information—as well as in highly heterogeneous single cell populations isolated from bulk tissue. We propose a specific metric ($ECDF_{|NES|}$) to assess whether a specific repertoire of interactomes is adequate for the metaVIPER analysis of an unknown or orphan tissue.

MetaVIPER is especially useful for the study of single cell biology, as its results are largely independent of sequencing depth and allow quantitative inference of protein activity even when the corresponding mRNA is undetectable. Indeed, differential activity of established lineage markers of T, B, and melanoma cells could be clearly assessed in single cells from a complex mixture, even though most of these markers were either not detected or could not be identified as statistically significantly differentially expressed at the mRNA level. The reduction in bias and batch effects is an additional advantage, allowing integration of datasets from multiple labs or generated at different times, thus addressing the important issue of single cell data reproducibility.

Among the most obvious limitations of the method, metaVIPER cannot accurately measure activity of proteins whose regulons are not adequately represented in at least one of the available interactomes. This includes proteins whose targets are exceedingly tissue-specific within rare tissue types and single cell sub-populations, for instance in LIHC and TGCT. As more interactomes are assembled, including by ARACNe analysis of single cell data from homogeneous sub-populations, this

limitation will be increasingly mitigated. This suggests that a concerted effort toward the generation of regulatory models representing distinct cellular compartments should be undertaken.

It should be noted that, while we used ARACNe as a methodology for interactome generation, there are many alternative/complementary methods to accomplish the same goal, ranging from DNA binding-site analysis[51,52], to correlation-based[53] and graphical-model-based[54], to literature-based approaches[55]. Comparison of VIPER performance using several of these methods was already discussed in ref. [7] and is thus not repeated here. In terms of the VIPER algorithm, as also discussed in ref. [7], alternative algorithms to transform a gene expression profile into a protein activity profile are still lacking but a thorough performance comparison can be easily performed once they become available. In general, the metaVIPER approach is independent of the specific algorithms used for either interactome reverse engineering or analysis and should thus be still fully applicable once VIPER alternatives emerge.

We have shown that VIPER-based elucidation of MR proteins using tissue lineage-specific interactomes can effectively identify reprogramming and pluripotency factors[13,21,22,56], as well as determinants of tumor states[11–13] and resistance to targeted therapy[17,36]. As a result, application of metaVIPER to single cell populations identified by cluster analysis could help identify critical determinants of lineage development, as well as distinct dependencies within molecularly heterogeneous sub-population in cancer tissues. For instance, it may help identify critical dependencies in chemoresistant cell niches, including rare tumor-initiating and tumor stem cell niches that have been shown to have poor sensitivity to standard chemotherapy and targeted therapy. Similarly, it could help identify drivers leading to aberrant reprogramming of physiologic cell states, such as recently reported in type II diabetes[57].

## Methods

**Regulatory networks**. All regulatory networks were reverse engineered by ARACNe[9] and summarized in Supplementary Table. Twenty-four core TCGA RNA-Seq derived interactomes are available in R-package aracne.networks from Bioconductor[24]. The TCGA human SKCM network was assembled from RNA-Seq profiles. TCGA RNA-Seq level 3 data (counts per gene) were obtained from the TCGSA data portal, and normalized by Variance Stabilization Transformation (VST), as implemented in the DESeq package from Bioconductor[58]. The human B lymphocyte interactome was reported by Basso et al.[9]. The human T lymphocyte interactome was reported by Piovan et al.[36]. The human brain tumor regulatory networks were assembled from four more gene expression datasets besides the TCGA glioblastoma RNA-Seq dataset. For the Rembrandt, Phillips[32], TCGA-Agilent, and TCGA-Affymetrix, informative probe clusters were assembled with the cleaner algorithm[59] and the expression data were summarized and normalized with the MAS5 algorithm, as implemented in the affy R-package from Bioconductor[60]. Differences in sample distributions were removed with the robust spline normalization procedure implemented in the lumi R-package from Bioconductor[61]. In a similar way, differences in sample distribution for the TCGA-Agilent dataset were removed by the robust spline normalization method. ARACNe was run with 100 bootstrap iterations using 1813 transcription factors (genes annotated in gene ontology molecular function database, as GO:0003700, "transcription factor activity", or as GO:0003677, "DNA binding", and GO:0030528, "transcription regulator activity", or as GO:0034677 and GO: 0045449, "regulation of transcription"), 969 transcriptional cofactors (a manually curated list, not overlapping with the transcription factor list, built upon genes annotated as GO:0003712, "transcription cofactor activity", or GO:0030528 or GO:0045449), and 3370 signaling pathway related genes (annotated in GO biological process database as GO:0007165 "signal transduction" and in GO cellular component database as GO:0005622, "intracellular", or GO:0005886, "plasma membrane"). Parameters were set to zero DPI (Data Processing Inequality) tolerance and MI (Mutual Information) $p$-value (using MI computed by permuting the original dataset as null model) threshold of $10^{-8}$.

**Associating somatic mutations with metaVIPER inference**. We consider somatic mutations that happen in the same amino acid of a protein within at least three patients as recurrent somatic mutations. Then for each protein, we did enrichment analysis with activity profile for each patient as signature, and patient harboring recurrent somatic mutation for that specific protein as enriching set. We consider proteins with significant enrichment score ($p < 0.01$) as showing significant association between inferred protein activity and recurrent somatic mutations. Then we checked the fraction of proteins that can be associated with recurrent somatic mutations, and used that as criteria in evaluating the performance between VIPER and metaVIPER. In order to get enough mutated patient samples for each protein, this analysis is done in a tumor type non-specific manner.

**Preparation of glioblastoma mouse model**. PDGFB–IRES–CRE expressing retrovirus was injected into the rostral subcortical white matter of adult $Pten^{lox/lox}/p53^{lox/lox}/luciferase-^{stop-lox}$ transgenic mice[30,31]. Mice developed brain tumors with the histopathological features of glioblastoma by 28 days post injection with retrovirus.

**Generating scRNA-Seq profiles for glioblastoma mouse model**. Following IACUC guidelines, animals were sacrificed at the first sign of morbidity. Ex vivo gross total resection of the tumor was performed and tumor cells were isolated using enzymatic digestion[62]. The isolated cells were cultured in a 2:1 ratio of basal media (DMEM, N2, T3, 0.5% FBS, and penicillin/streptomycin/amphotericin) in B104 conditioned media[63]. This media was further supplemented with PDGF–AA (Sigma-Aldrich; St. Louis, MO) and FGFb (Gibco; Grand Island, NY) to a concentration of 10 ng/ml. We then obtained 85 scRNA-Seq profiles using the Fluidigm C1 system. We loaded dissociated cells into a Fluidigm Integrated Fluidic Circuit with capture sites designed for 10–17 μm diameter cells after staining the single cell suspension with Calcein AM (Life Technologies). We then imaged the cells that had been captured on-chip with both bright field and fluorescence microscopy using an inverted Nikon Eclipse Ti–U epifluorescence microscope with a ×20, 0.75 NA air objective (Plan Apo λ, Nikon), a 473 nm diode laser (Dragon Lasers), and an electron multiplying charge coupled device (EMCCD) camera (iXON3, Andor Technologies). This allowed us to identify capture sites with zero, one, and more than one cell and also to identify capture sites containing living cells, based on the Calcein AM fluorescence. We then lysed the cells, reverse transcribed mRNA into cDNA, and pre-amplified full-length cDNA by PCR automatically using the Fluidigm C1 Autoprep instrument according to the manufacturer's instructions. Finally, we harvested individual cDNA libraries from the microfluidic device and converted them into indexed, Illumina sequencing libraries by in vitro transposition, and PCR using the Nextera system (Illumina). The pooled libraries were sequenced on a single lane of an Illumina HiSeq 2000 with single-end 100-bp reads. After demultiplexing, the resulting raw reads were aligned to the murine genome and transcriptome annotation (mm10, UCSC annotation from Illumina iGenomes) with Tophat 2. Uniquely aligned, exonic reads were then quantified for each gene using HTSeq.

**Code availability**. metaVIPER is implemented in viper function from Bioconductor R-package VIPER: https://www.bioconductor.org/packages/release/bioc/html/viper.html. ARACNe algorithm: http://califano.c2b2.columbia.edu/aracne. Custom scripts will be provided upon request to the corresponding authors.

**Data availability**. scRNA-Seq data for the mouse glioblastoma model described in the paper have been deposited at the Gene Expression Omnibus (GEO) under accession number GSE95157. R-package aracne.networks is available on Bioconductor (10.18129/B9.bioc.aracne.networks). SKCM, B, and T lymphocyte interactomes (10.6084/m9.figshare.4833704). Brain tumor interactomes (10.6084/m9.figshare.4648765.v1). TCGA expression and somatic mutation profile: http://cancergenome.nih.gov/. REMBRANDT data set: https://gdoc.georgetown.edu/gdoc/. COSMIC somatic mutation profile: http://cancer.sanger.ac.uk/cosmic. Filtered PBMC scRNA-Seq expression profiles generated using 10× Genomics V2 chemistry: https://support.10xgenomics.com/single-cell-gene-expression/datasets/2.0.1/pbmc4k. Filtered PBMC scRNA-Seq expression profiles generated using 10× Genomics V1 chemistry: https://support.10xgenomics.com/single-cell-gene-expression/datasets/1.1.0/pbmc3k. All relevant data are available from the authors.

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

## Acknowledgements

This work was supported by US National Institutes of Health grants R35 CA197745-03 and U01 CA217858.

## Author contributions

A.C. and M.J.A. conceived and initiated the project. H.D., M.J.A., and E.F.D. performed the analysis. A.M.S., A.M., and P.D.C. prepared GBM mouse model. S.B., C.G., and P.A.S. generated scRNA-Seq profiles. H.D., M.J.A., and A.C. prepared the manuscript.

**Additional information**

**Competing interests:** M.J.A. is chief scientific officer of DarwinHealth Inc. A.C. is founder and equity holder of DarwinHealth Inc., a company that has licensed some of the algorithms used in this manuscript from Columbia University. Columbia University is also an equity holder in DarwinHealth Inc. The remaining authors declare no competing interests.

