## [Peer Review File · Nature Communications]

Reviewers' comments:

Reviewer #1 (Remarks to the Author):

This paper presents a method that is an extension of ARACNe and VIPER, both were developed by the same group. ARACNe constructs gene regulation network through coexpression analysis, VIPER estimates the activity of a regulatory protein in a certain cell-type, whereas the new method presented here, called metaVIPER, intends to estimate protein activities without prior knowledge of tissue information. This could happen, for example, in cancer tissues as well as in single-cell analysis. The main strategy is to first derive a reference set of regulons in well-defined tissue types, and then compare each regulon with the differentially expressed genes and select the one that has strongest overlap. Interestingly, this approach works well in identifying unknown tissue types from cancer samples and single-cell analysis.

This work is very interesting. Reconstructing gene regulation networks in poorly characterized tissues such as cancer is a main challenge. The method proposed here seems to be a powerful tool for such analysis. The paper is very well written and would be of interest to the broad community.

A few minor comments:

1. It seems that the method would work only if there exist a diverse pool of tissues for which regulons can be well-characterized. Is there a guide on how many such tissues are needed? Presumably it would also depend on the variability of the regulatory protein of interest?
2. The single-cell analysis is very interesting, but I am a bit surprised to see that closely related cell populations (as judged by the gene expression patterns) can be easily distinguished based on regulons obtained from tissue-level analysis, which presumably contains heterogeneous cell types. It would be great if the authors could provide some intuitive explanation.

Reviewer #2 (Remarks to the Author):

From a technical point of view this work is sound and the paper is also mostly well-written (some minor criticisms below).

The essence of this paper is the observation, which others have made use of before, that pooling data across diverse expression samples from different tissues is more informative of cell regulation than only closely-related samples. This observation is applied to improve the authors' existing method, VIPER which in turn builds on their much older very well-cited method ARACNe. The outcome is that this new extension improves performance, and in particular overcomes certain requirements imposed by the original approach. Users of VIPER will be excited by this advance, and it will also open up use to new users with different data requirements.

The introduction fails to put the work in context within the field. Would the authors have the reader believe they are the only group in the field? The authors only cite their own work in relation to the methodology (for which there is plenty of work to compare it to), the application (fewer, but still important citations to be made) and do not benchmark against other methods. Their previous papers have compared more adequately to other methods in their benchmarks, and have shown more direct results. The benchmarking is only a proxy for their stated end goal of identifying master regulators. The paper could be improved by either experimentally validating the master regulators directly, re-stating the objective to match the benchmark, or (probably) admitting the

gap between the stated goal and the benchmark.

Something which I would have really liked to have seen, and which would really add to the paper (but should not be a barrier to publication) is the performance of MetaVIPER on samples from different sources. Single cell sequence data, something which the paper makes much interest of, is notoriously difficult to use from different sources, and it would be of great interest to know if MetaVIPER can attain good results across different single cell (and other) sequence data sources where others have failed.

The remaining remarks are minor, and refer to the writing.

Search for the following grammatical errors in the text:

"... targets of a specific proteins in ..."

"... a number of distinct benchmark based on ..."

"... identify classical targets targeted therapy."

"... systematically assess method's performance in orphan tissues ..."

"Quality of a single cell gene expression profiles is ..."

The discussion uses too many sensational words without scientific justification. "comprehensive, systematic" please explain what makes it comprehensive and systematic. "Critically ... " why is it critical? Critical for what?etc. Statements need to be backed up (e.g. with reference to the main body of the paper).

The discussion uses the word dissected without explaining it. The discussion implies all specialized tissues correlate well with a analysis including a matched interactome when excluded, whereas there is only a single example tested. This is misleading. Also that sentence is nonsensical as is, although from reading the rest of the paper it is clear what the intended meaning is. The word 'mitigated' is not really the correct term in places where it is used. It should be written that the authors have 'previously' shown things with VIPER to distinguish from work done in this paper.

Reviewer #1 (Remarks to the Author):

This paper presents a method that is an extension of ARACNe and VIPER, both were developed by the same group. ARACNe constructs gene regulation network through coexpression analysis, VIPER estimates the activity of a regulatory protein in a certain cell-type, whereas the new method presented here, called metaVIPER, intends to estimate protein activities without prior knowledge of tissue information. This could happen, for example, in cancer tissues as well as in single-cell analysis. The main strategy is to first derive a reference set of regulons in well-defined tissue types, and then compare each regulon with the differentially expressed genes and select the one that has strongest overlap. Interestingly, this approach works well in identifying unknown tissue types from cancer samples and single-cell analysis.

This work is very interesting. Reconstructing gene regulation networks in poorly characterized tissues such as cancer is a main challenge. The method proposed here seems to be a powerful tool for such analysis. The paper is very well written and would be of interest to the broad community.

A few minor comments:

1. It seems that the method would work only if there exist a diverse pool of tissues for which regulons can be well-characterized. Is there a guide on how many such tissues are needed? Presumably it would also depend on the variability of the regulatory protein of interest?

This is a very relevant point and the answer is not completely straightforward, because the number of required interactomes depends, of course, on the specific protein being evaluated, on the degree of conservation of its regulon across tissues, and on which tissues are currently represented in the collection of metaVIPER interactomes. Clearly, the more interactomes are available, the more likely it will be that a protein's regulon in the unknown tissue of interest will be replicated, at least in part, in one of the tissue context for which an interactome has already been assembled. But can this be formulated in quantitative fashion?

To address this important question in a more thorough and quantitative fashion, we now propose a metric representing the overall normalized enrichment score of all proteins based on their differentially expressed targets within the specific (and possibly unknown) context of interest. This is based on the fact that, as discussed in the manuscript (see Supplementary Fig. 1), regulons that are not representative of the specific tissue biology will not produce any statistically-significant enrichment. A metric can thus be computed as the *Empirical Cumulative Distribution Function of the absolute value of the VIPER Normalized Enrichment Score* ($ECDF_{|NES|}$) of all proteins in an orphan tissue sample or samples. In a new Supplementary Fig. 7, we show the range of the $ECDF_{|NES|}$ as well as the standard deviation, using a violin plot representation, across all cohorts and corresponding tissue-matched interactomes in TCGA, as well as the average of the $ECDF_{|NES|}$ across all tissues. Thus, we can use the lowest value across all analyzed tissue types as a minimum threshold to determine whether a new, unknown tissue is well resolved by metaVIPER using the current set of interactomes or whether additional interactomes are required. As an example, we show the case of a hematopoietic cancer studied with a GBM-Specific interactome. As shown, the $ECDF_{|NES|}$ in the hematopoietic tumor cohort is significantly below that of

the lowest tissue-matched interactome, thus suggesting that metaVIPER cannot properly resolve this context using only the GBM interactome.

We revised the manuscript to reflect this point.

2. The single-cell analysis is very interesting, but I am a bit surprised to see that closely related cell populations (as judged by the gene expression patterns) can be easily distinguished based on regulons obtained from tissue-level analysis, which presumably contains heterogeneous cell types. It would be great if the authors could provide some intuitive explanation.

We understand the concern by the reviewer but this is likely due to a common misunderstanding. Specifically, cells can be in different states without necessarily having a different regulatory network. For instance, as we reported for GBM in (Carro, MS, *et al.* "The transcriptional network for mesenchymal transformation of brain tumors." *Nature* 463.7279 (2010): 318.), we could identify master regulators for three different subtypes of GBM using a network that is inferred by bulk tissues representing not only different subtypes of the disease but likely also heterogeneous mixture of multiple cell types. In general, we have shown that if the lineage of the cells is conserved, then the differences in the underlying regulatory model are minimal (5% across 18 different types of normal and lymphoma related B cells). Even when the models are different, there will be some regulons that are well conserved between multiple tissue types and these will contribute to providing the ability to distinguish different subtypes. Of course, the optimal way to use metaVIPER would thus be to first identify sub-populations, then using ARACNe to generate sub-population specific networks, and then re-analyze the data to get even better ability to perform tissue context specific assessment of protein activity.

To specifically address the reviewer's question, the reason why MES and PN subtypes can be distinguished by using GBM tissue-level networks is essentially that the underlying regulatory network of the MES and PN subtypes are highly overlapping. Differences in MES vs. PN state are caused by different activity of a handful of MR proteins rather than by different underlying regulatory networks.

Reviewer #2 (Remarks to the Author):

From a technical point of view this work is sound and the paper is also mostly well-written (some minor criticisms below).

The essence of this paper is the observation, which others have made use of before, that pooling data across diverse expression samples from different tissues is more informative of cell regulation than only closely-related samples. This observation is applied to improve the authors' existing method, VIPER which in turn builds on their much older very well-cited method ARACNe. The outcome is that this new extension improves performance, and in particular overcomes certain requirements imposed by the original approach. Users of VIPER will be excited by this advance, and it will also open up use to new users with different data requirements.

The introduction fails to put the work in context within the field. Would the authors have the reader believe they are the only group in the field? The authors only cite their own work in relation to the

methodology (for which there is plenty of work to compare it to), the application (fewer, but still important citations to be made) and do not benchmark against other methods. Their previous papers have compared more adequately to other methods in their benchmarks, and have shown more direct results. The benchmarking is only a proxy for their stated end goal of identifying master regulators. The paper could be improved by either experimentally validating the master regulators directly, restating the objective to match the benchmark, or (probably) admitting the gap between the stated goal and the benchmark.

We really appreciate the comments of the reviewer and apologize if the manuscript generates the impression that we are the only group working in this field. However, we had a similar discussion with the Nature Genetics reviewers and editor and we agreed that, while there are multiple labs that have developed and used reverse engineering algorithm and network based algorithm to study cellular behavior, there really are no alternative algorithms to VIPER that we know of or that the reviewers could identify in terms of transforming a gene expression profile into a protein activity profile. As discussed in the VIPER manuscript (Alvarez, Mariano J., *et al.* "Functional characterization of somatic mutations in cancer using network-based inference of protein activity." *Nature genetics* 48.8 (2016): 838-847.), VIPER can leverage any transcriptional network reverse engineering algorithm. For that step, the choice of ARACNe is only one of many. Indeed, in that manuscript we did perform a comparative VIPER analysis when using different methods for mapping regulatory networks. As a result, we are not sure how to perform a comparative analysis as requested by the reviewer.

With respect to experimental validation, we have published dozens of manuscripts where predictions by VIPER and by its predecessor, MARINa, were experimentally validated. In this manuscript, we show that predictions made by metaVIPER, when eliminating the matched interactome from the analysis, are virtually identical to those made using the tissue-matched interactomes. In addition, we show metaVIPER produces protein activity estimations that co-segregates better with mutational data. As a result, we do not see how providing validation for one or two MRs would be more convincing than the wealth of data that has already been produced, see (A. Califano, M. J. Alvarez, "The recurrent architecture of tumor initiation, progression and drug sensitivity". *Nat Rev Cancer* 17, 116-130, 2017) for a comprehensive compendium of all these studies.

Something which I would have really liked to have seen, and which would really add to the paper (but should not be a barrier to publication) is the performance of MetaVIPER on samples from different sources. Single cell sequence data, something which the paper makes much interest of, is notoriously difficult to use from different sources, and it would be of great interest to know if MetaVIPER can attain good results across different single cell (and other) sequence data sources where others have failed.

This is a very good point. To address this reviewer's question, we analyzed filtered PBMC scRNA-Seq data generated using V2 (<https://support.10xgenomics.com/single-cell-gene-expression/datasets/2.0.1/pbmc4k>, most updated) and V1 (<https://support.10xgenomics.com/single-cell-gene-expression/datasets/2.0.1/pbmc4k>, most updated) and V1 (<https://support.10xgenomics.com/single-cell-gene-expression/datasets/2.0.1/pbmc4k>, most updated).

cell-gene-expression/datasets/1.1.0/pbmc3k) chemistry from 10x Genomics. While the gene expression profile data shows a strong batch effect in t-SNE projections, this was virtually absent in metaVIPER-inferred protein activity. We provide these results as one additional supplementary figure, Supplementary Figure 6.

We revised the manuscript according to the comment.

The remaining remarks are minor, and refer to the writing.

Search for the following grammatical errors in the text:

"... targets of a specific proteins in ..."

"... a number of distinct benchmark based on ..."

"... identify classical targets targeted therapy."

"... systematically assess method's performance in orphan tissues ..."

"Quality of a single cell gene expression profiles is ..."

We thank the reviewer for these recommendations and we have addressed all of these in the revised manuscript.

The discussion uses too many sensational words without scientific justification. "comprehensive, systematic" please explain what makes it comprehensive and systematic. "Critically ... " why is it critical? Critical for what?etc. Statements need to be backed up (e.g. with reference to the main body of the paper).

The discussion uses the word dissected without explaining it. The discussion implies all specialized tissues correlate well with a analysis including a matched interactome when excluded, whereas there is only a single example tested. This is misleading. Also that sentence is nonsensical as is, although from reading the rest of the paper it is clear what the intended meaning is. The word 'mitigated' is not really the correct term in places where it is used. It should be written that the authors have 'previously' shown things with VIPER to distinguish from work done in this paper.

We are sorry if the use of words appeared sensationalistic. However, we are not sure we understand how using the term systematic may appear sensationalistic. By systematic (or comprehensive for that matter), we simply mean that these analyses can be performed identically, using consistent statistical criteria for each one of 6,000 proteins for which regulons are available and across any tissue of interest. This is not the case for other methodologies (e.g. antibody based or Mass-spec based) aimed at measuring protein abundance or activity, which can only be used on a very small subset of proteins or only when large amount of tissue is available.

In any case, we revised the manuscript according to reviewer's comments to avoid any appearance of sensationalistic statements and to clarify the use of specific words.

Reviewers' comments:

Reviewer #1 (Remarks to the Author):

The authors have done a great job revising their manuscript and completely addressed the concerns in my original review. The Empirical Cumulative Distribution Function metric added in the revision is very interesting and may serve a very powerful guide for research design.

Reviewer #2 (Remarks to the Author):

In re-reviewing this paper it has become clear that I did not understand some parts of it during the first review. It was by reading the authors' other papers that many things became clear that were not explained in this manuscript. If this paper is going to be published for a non-specialist audience then the authors should not assume that the readers have already read their other papers, especially the VIPER paper. For example 'protein activity' is not introduced (a reader could infer differential expression, or protein abundance), and a precise meaning of which in this context is completely central to the paper. A quick read of the VIPER paper cleared this up easily, but this paper in general does not stand alone, and the abstract and introduction would lead the reader to believe it is about master regulators, which never come into it in the end. I apologise for some of this comment being partly new since the first review, but I did touch on the gap between the stated goal of master regulators and the actual work on protein activity. This comment still stands.

Why was Meta-VIPER removed from the title?

The term NES is not defined.

The authors dealt with most other comments from the first review, but the possibility of benchmarking is still outstanding (the Nature Genetics editor's opinion notwithstanding). Whilst there may not be an identical approach to the authors', the better differential expression methods would be a very standard thing to compare to on their benchmarks. Since their approach is acting as a proxy, it would be important to know how it performs relative to differential expression which would be another simpler proxy that before metaVIPER people would have employed. Especially with regard to metaVIPER there are some differential expression and expression clustering algorithms that specifically attempt to cater for situations of low numbers of replicates in samples by using information across other samples for which there are more replicates. This is exactly what metaVIPER does for VIPER, so a direct comparison is warranted.

Reviewer #2 (Remarks to the Author):

In re-reviewing this paper it has become clear that I did not understand some parts of it during the first review. It was by reading the authors' other papers that many things became clear that were not explained in this manuscript. If this paper is going to be published for a non-specialist audience then the authors should not assume that the readers have already read their other papers, especially the VIPER paper. For example 'protein activity' is not introduced (a reader could infer differential expression, or protein abundance), and a precise meaning of which in this context is completely central to the paper. A quick read of the VIPER paper cleared this up easily, but this paper in general does not stand alone, and the abstract and introduction would lead the reader to believe it is about master regulators, which never come into it in the end. I apologise for some of this comment being partly new since the first review, but I did touch on the gap between the stated goal of master regulators and the actual work on protein activity. This comment still stands.

We agree with the reviewer that the manuscript, in its current form, is not self-contained and relies heavily on the readers being familiar with the VIPER algorithm. To address this issue, we have modified the manuscript to explain the basis for using the expression of a protein's transcriptional targets (direct for TFs, and least indirect for signaling proteins) as a multiplexed gene reporter assay to assess its activity. We thank the reviewer for this comment as sometimes, one may take things for granted and this may be confusing to a general audience.

Why was Meta-VIPER removed from the title?

Unfortunately, the journal has specific requirements about the title not containing punctuation marks. Therefore, we had removed the "metaVIPER:" element. However, depending on the editor's and reviewers' recommendations, we would be happy to change the title to "Quantitative Assessment of Protein Activity in Orphan Tissues and Single Cells Using the metaVIPER Algorithm"

The term NES is not defined.

In the revised manuscript, we defined NES as “Normalized Enrichment Score”, which represents the Kolmogorov-Smirnov statistics to measure the enrichment of a protein transcriptional targets in differentially expressed genes.

The authors dealt with most other comments from the first review, but the possibility of benchmarking is still outstanding (the Nature Genetics editor's opinion notwithstanding). Whilst there may not be an identical approach to the authors', the better differential expression methods would be a very standard thing to compare to on their benchmarks. Since their approach is acting as a proxy, it would be important to know how it performs relative to differential expression which would be another simpler proxy that before metaVIPER people would have employed. Especially with regard to metaVIPER there are some differential expression and expression clustering algorithms that specifically attempt to cater for situations of low numbers of replicates in samples by using information across other samples for which there are more replicates. This is exactly what metaVIPER does for VIPER, so a direct comparison is warranted.

We apologize if our comments about the algorithm comparison in Nature Genetics may have appeared smug. This was not our intention at all. We were simply stating that it is difficult to compare to other methodologies because we are not aware of any other published algorithm to infer protein activities from gene expression data. However, we completely understand the reviewer’s desire to at least compare metaVIPER to gene expression based analyses. To address this request, we now introduce a systematic benchmarking of single cell analyses using gene expression and protein activity. We thank the reviewer for this suggestion as it further highlights the advantages of metaVIPER analysis, especially in the context of single cell data, as shown in a new Figure 4 that summarizes these results.

Specifically, one of the critical issues in using single cell gene expression is that the majority of genes produce no detectable mRNA reads. In contrast, metaVIPER can estimate activity of 6,000 critical proteins, including all TFs, co-TFs, signaling proteins, and chromatin remodeling enzymes, even if their expression is undetectable. To systematically assess this difference, we analyzed single cell data from a mixture of T cells, B cells, and melanoma cells (Ref. 35) either by gene expression analysis or by metaVIPER analysis. We first clustered single cells in individual sub-populations – by expression or activity clustering (Fig. 4a and 4f, respectively) – and then assessed the top 100 most differentially expressed genes and most differentially active proteins in each pairwise comparison (i.e., T cells vs. melanoma, T cells vs. B cells, and B cells vs. melanoma) by using a “virtual bulk” tissue expression profile obtained by summing the mRNA counts for each gene across all cells in a cluster. Finally, we assessed how many of these differentially genes/proteins could be recapitulated as differentially expressed/active at the single cell level (e.g., in an individual T cell vs. the bulk of melanoma cells). Figure 4b shows that at the single gene level metaVIPER systematically outperforms gene expression analysis. This difference is further exacerbated when considering a pair of genes/proteins (Fig. 4g).

This difference allows us to perform the equivalent of FACS sorting on any pair of relevant markers chosen from the 6,000 proteins assessed by metaVIPER (Fig4h – j). The same analysis is virtually impossible at the gene expression level, as the majority of single cells would lay either on the y-axis (i.e. no mRNA reads for the first marker) or on the x-axis (i.e. no mRNA reads for the second marker), with a substantial number of cells actually being at the intersection of the two axes ($x = 0$; $y = 0$) (Fig4c – e). In

For instance, consider key lineage markers, such as CD19 and CD3, which are expressed in B and T cells respectively. From metaVIPER analysis (Fig. 4j), one can clearly observe a CD19+/CD3- cluster corresponding to B cells, a CD19-/CD3+ cluster corresponding to T cells, and a CD19-/CD3- cluster corresponding to melanoma cells. In contrast, no clusters can be detected by gene expression analysis, with only a handful of single cells having measurable expression for both marker (Fig. 4e). This allows us to perform “virtual FACS” analysis across any pair of proteins assessed by metaVIPER, as shown by additional examples, such as FOXP3, vs. PAX5 and POU2F2 vs. STAT4. This also shows that metaVIPER can be used to probe into the biology of individual cells based on the activity of critical lineage markers (TFs/co-TFS) and surface markers (signaling proteins), whose encoding genes are undetectable at the mRNA level.

To address these questions, we updated our ARACNe network collection to include interactomes reconstructed from 4 GBM datasets (microarray-based), as well as B-cell, T-cell and melanoma (SKCM) datasets. We updated the single cell analysis accordingly. Updates include:

1. ARACNe network collection, including description in Supplementary Table, and data repository in Figshare.
2. Figure 2, 3; Supplementary Figure 3, 4

Reviewers' comments:

Reviewer #2 (Remarks to the Author):

The authors have addressed everything apart from the issue, critical to a methods paper, if benchmarking against independent 3rd party software. The authors have compared their software to their own expression analysis, but this is not what I suggested.

I suggested that their approach be compared to the current best in the field for their application case. I also explicitly suggested that the authors find a method that uses information across multiple samples to bootstrap missing information which is the principle they have followed. I am not up-to-date on what is the best in the field, especially for single-cell data, but there are popular methods such as EdgeR and DESeq, with many more listed here:

https://en.wikipedia.org/wiki/List_of_RNA-Seq_bioinformatics_tools#Co-expression_networks

A search reveals DGEClust which uses the same principle as the authors' approach of using information across samples, but there's probably a better/newer more popular method that could be used.

I leave it to the authors to select and justify the choice of 3rd party software for comparison.

As per our prior discussions, comparative analysis of metaVIPER performance vs. other algorithms is challenging. This is because most of the methodologies developed to deal with the low-depth profiling of single cell transcriptomes and consequent high gene dropout rates – e.g. Vu et al., 2016 PMID: 27153638, Kharchenko et al., 2014 PMID: 24836921 or Finak et al., 2015 PMID: 26653891 – were not designed to reduce dropout effects at the single cell level but only at the sub-cluster level. I.e., they are meant to improve differential expression analysis between distinct sub-populations of molecularly similar cells (clusters) rather than between individual cells. By dropout genes we indicate genes having zero reads in a specific scRNA-Seq profile.

[Redacted]

We also systematically benchmarked metaVIPER's performance against the best methodology for gene expression profile normalization, i.e., the RSEM algorithm (Li et al., 2011 PMID: 21816040). RSEM pre-assembles sequencing reads into transcripts, thus resulting in a more accurate gene expression assessment.

We apologize the fact that the graph used to assess the latter comparison (metaVIPER vs. RSEM) in the previous resubmission may have been misleading. Indeed, for the RSEM analysis in that graph, we had considered only cells where at least one read from the gene of interest was detected (i.e. dropout cells for each tested gene were not included in the analysis), thus significantly improving RSEM results. To address this problem, we modified the graph (see below) to show the analysis of all single cell pairs, such that the number of single cell pairs analyzed is the same for metaVIPER, [Redacted], and RSEM. As shown, the difference in reproducibility is remarkable with a mode of 80% of cell pairs showing statistically reproducible differential activity of the top 100 most differentially active proteins in the corresponding sub-population bulk profiles (metaVIPER, cyan curve) vs. only 15% and 10% for the RSEM-based (magenta curve) and [Redacted] (yellow curve) analysis of the top 100 differentially expressed genes in the same single cell pair, respectively (panel C). The difference becomes even more striking when a pair of genes/proteins is considered (panel D), as would be the case for virtual FACS analyses.

[Redacted]

Results:

To address the remaining reviewer's concerns, we have modified the manuscript (changes tracked in red) to include the results of the more comprehensive RSEM vs. metaVIPER benchmark analysis, including changes to the content and legend of Figure 4. We also report the full comparison of metaVIPER to both RSEM and [Redacted] in the rebuttal for the reviewer's consideration.

Figure 4: Comparative analysis of single cell metaVIPER performance compared to gene expression based methods.

We identified the 100 most differentially expressed genes and differentially active proteins based on the analysis of 5 synthetic bulk samples created by averaging the expression of 100 randomly selected single cells from the melanoma, B cell, and T cell population clusters, respectively. (A, B) Based on t-SNE analysis, synthetic bulk samples clustered more tightly when analyzed based on VIPER-inferred protein activity than based on gene expression. (C). This panel shows the percent of the top 100 most differentially expressed genes/active proteins (Y axis) recapitulated as significantly differentially expressed/active in a given fraction of individual cells from two different clusters (e.g., T cells vs B cells) (X axis). The yellow, magenta, and turquoise curves show the results of [Redacted], RSEM, and metaVIPER-based analyses, respectively. (D). The same analyses were repeated to assess reproducible differential expression/activity of a gene/protein pair, as relevant for virtual FACS analyses. (F-H) Virtual FACS analyses using expression and activity of established lineage marker TFs by RSEM, [Redacted], and metaVIPER analysis (see main text and **Figure 3** for details). (I-K) Virtual FACS analysis using expression and activity of *STAT4* and *POU2F* – both identified as differentially expressed and active candidate biomarkers from bulk sample analyses, – using the same three methods. (L-N) Virtual FACS analysis based on expression and activity of CD3 and CD19 cell surface markers, as used in standard FACS analyses, using the same three methods. We also provide an example showing how B and T cells would be separated in a *bona fide* experimental FACS analysis using CD3 and CD19 antibodies (E, figure from <https://www.tonbobio.com/antibodies-and-reagents/percp-anti-human-cd19-sj25c1.html>)

REVIEWERS' COMMENTS:

Reviewer #2 (Remarks to the Author):

The benchmarking has now been addressed. Benchmarking is often challenging, especially when doing something different or new, but comparisons, even if not direct, are still important.